# Neurturin regulates the lung-resident macrophage inflammatory response to viral infection

Emma Connolly[1,2] , David J Morgan[1,2], Miriam Franklin[1,2], Angela Simpson[3], Rajesh Shah[4], Oliver J Brand[1,2], Christopher P Jagger[1,2], Joshua Casulli[1,2], Karishma Mohamed[1,2], Aleksander M Grabiec[2] , Tracy Hussell[1,2]

**Lung-resident macrophages are crucial to the maintenance of health and in the defence against lower respiratory tract infections. Macrophages adapt to local environmental cues that drive their appropriate function; however, this is often dysregulated in many inflammatory lung pathologies. In mucosal tissues, neuro-immune interactions enable quick and efficient inflammatory responses to pathogenic threats. Although a number of factors that influence the antimicrobial response of lung macrophages are known, the role of neuronal factors is less well understood. Here, we show an intricate circuit involving the neurotrophic factor, neurturin (NRTN) on human lung macrophages that dampens pro-inflammatory cytokine release and modulates the type of matrix metalloproteinases produced in response to viral stimuli. This circuit involves type 1 interferon–induced up-regulation of RET that when combined with the glial cell line-derived neurotrophic factor (GDNF) receptor $\alpha$2 (GFR$\alpha$2) allows binding to epithelial-derived NRTN. Our research highlights a non-neuronal immunomodulatory role for NRTN and a novel process leading to a specific antimicrobial immune response by human lung-resident macrophages.**

## Introduction

Lower respiratory tract infections and chronic respiratory diseases are within the top four causes of mortality globally (Barnes, 2019). Therefore, understanding the inflammatory cascades involved in the pathogenesis of these diseases is critical in developing new therapeutics and alleviating the clinical burden. Because of their location in the airways, lung-resident alveolar macrophages are in direct contact with the external environment and are at the first line of defence against infectious agents. For this reason, macrophage function needs to be tightly regulated to maintain homeostasis through tolerance to cellular debris and innocuous antigens, while still mounting an effective immune response against harmful pathogens and limiting bystander tissue damage (Hussell & Bell, 2014). Upon host infection with a virus, lung macrophages produce a range of cytokines, including type I IFNs, IL-6, and IL-12, in response to TLR activation (Yan & Chen, 2012; Goritzka et al, 2015). Type I IFNs, in turn, trigger the JAK—signal transducers and activators of transcription (STAT) pathway, which leads to the activation of a large number of interferon stimulated genes (Ivashkiv & Donlin, 2014). These genes encode proteins that induce cell-intrinsic anti-viral defences, but also cytokines and chemokines that are critical in the immune response to clear the threat (Rauch et al, 2013; Rusinova et al, 2013). However, the specific interferon-stimulated genes induced in macrophages by a given pathogen or TLR receptor and how this directs the ensuing inflammatory cascade, especially in humans, is largely unknown.

Mucosal sites, including the lung, are highly innervated with peripheral nerves and growing evidence implicates neuro-immune interactions in health and disease in the gastrointestinal tract (Chesne et al, 2019). Conversely, how neuronal signals affect immune cell function in the lung is less well established. RET is a receptor tyrosine kinase that is activated by glial cell line–derived neurotrophic factor (GDNF) family receptors and their ligands (GFLs). GDNF binds to GDNF family receptor (GFR)$\alpha$1, neurturin (NRTN) to GFR$\alpha$2, artemin to GFR$\alpha$3, and persephin to GFR$\alpha$4 (Airaksinen & Saarma, 2002). Ligand binding to the receptor promotes dimerization of RET and subsequent changes in biological processes such as differentiation, proliferation, and apoptosis (Lemmon & Schlessinger, 2010), predominantly characterised in the nervous system. RET is essential for the development of the enteric nervous system (Schuchardt et al, 1994) and, through the formation of a GFR$\alpha$3/Artemin/RET signalling complex, governs the formation of enteric Peyer's patches (Veiga-Fernandes et al, 2007). However, the GDNF family also have non-neuronal immune-modulatory functions in the gastrointestinal tract. In this setting, TLR agonists or alarmins stimulate enteric glial cells to produce the ligands, GDNF, and NRTN (Ibiza et al, 2016). This in turn leads to the activation of RET-expressing

---

[1]The Lydia Becker Institute for Immunology and Inflammation, The University of Manchester, Manchester, UK  [2]Manchester Collaborative Centre for Inflammation Research, The University of Manchester, Manchester, UK  [3]Division of Infection, Immunity and Respiratory Medicine, Manchester Academic Health Sciences Centre, University of Manchester, Manchester, UK  [4]Department of Thoracic Surgery, University Hospital of South Manchester, Manchester, UK

Correspondence: tracy.hussell@manchester.ac.uk
Aleksander M Grabiec's present address is Department of Microbiology, Faculty of Biochemistry, Biophysics and Biotechnology, Jagiellonian University, Kraków, Poland

group 3 innate lymphoid cells (ILC3s) and subsequent production of IL-22 in a STAT3-dependent manner (Ibiza et al, 2016). IL-22 then acts on gut epithelial cells to drive antimicrobial responses, therefore implicating the GDNF family in promoting host defence against infection (Ibiza et al, 2016).

Evidence also implicates the GDNF family in other diverse aspects of the immune system. NRTN exhibits anti-inflammatory properties by inhibiting the secretion of IL-6 and TNFα from LPS-stimulated human PBMCs (Vargas-Leal et al, 2005) and dampens the inflammatory response to allergens in a mouse model of asthma (Mauffray et al, 2015). Within the bone marrow, RET signalling promotes haematopoietic stem cell survival and function (Fonseca-Pereira et al, 2014) and regulates their daily migration into the circulation (Garcia–Garcia et al, 2019). Meanwhile GFRα2 has been defined as a marker for the human intermediate monocyte subset (Wong et al, 2011) and B cell progenitors (Jensen et al, 2018). Therefore, this accumulating data clearly shows that GDNF family members play significant roles in shaping immune responses.

We hypothesised that receptors classically attributed to neuronal cells would be expressed by lung macrophages and drive their function. We searched for neuronal receptors on human lung-resident macrophages and found a specific sequence of steps leading to formation of a GFRα2-RET-NRTN complex that was amplified in lung macrophages from chronic smokers and tumour tissue. We determined that viral infection, or mimics thereof, specifically enhanced the production of the GFRα2 ligand, NRTN, from lung epithelium, whereas type I IFNs induced expression of the signalling receptor RET in lung macrophages. RET activation led to a reduction in pro-inflammatory cytokine release and a switch in the type of matrix metalloproteinases (MMPs) produced from macrophages. This highly specific outcome following the precise development of a multi-component receptor highlights an important, non-neuronal role for GFRα2 in directing macrophage function to promote host defence.

# Results

### Human lung macrophages constitutively express GFRα2 and the signalling co-receptor RET is induced by TLR activation

Though neuronal factors have been predominantly characterised in the nervous system, we observed high GFRα2 expression under conditions of GM-CSF, compared with M-CSF, differentiation of murine BMDMs by $RT^2$ profiler array (Fig S1A and B) and confirmed this by quantitative (q)PCR (Fig S1C and D). GMCSF plays a significant role in lung immune homeostasis and is critical in the development of alveolar macrophages (Schneider et al, 2014). Consistent with in vitro data, GFRα2 was highly expressed on mouse (Fig S1E) and human lung macrophages, compared with other members of the GDNF family, at the mRNA level (Fig 1A). GFRα2 was also expressed on human peripheral blood monocytes and human monocyte-derived macrophages (MDMs) (Fig 1B). On the other hand, the co-receptor, RET, could not be detected in monocytes, human MDMs or human lung macrophages (MΦs) at steady state (Fig 1A and B). GFRα2 and RET expression was further confirmed at the

protein level, with GFRα2 expression observed in human lung macrophages and human MDMs (Fig 1C and D), whereas the co-receptor RET was only detected in PMA-treated THP-1 macrophages (Fig 1D). To determine the trigger for macrophage expression of RET; human MDMs were stimulated with Th1/Th2 cytokines and TLR agonists. Cytokines associated with polarising macrophages had little effect on RET expression (Fig 1E); however, RET mRNA expression was significantly induced by the TLR3 agonist, polyI:C, and to a lesser extent by the TLR4 and TLR7/8 agonists, LPS and R848, respectively (Fig 1E). Therefore, although macrophages constitutively express GFRα2, the signalling component, RET, is induced by stimuli associated with viral infection.

### Lung epithelial cells express the GDNF family ligand NRTN

We next determined that one possible source of the GFRα2 ligand was airway epithelial cells, where NRTN was detected under basal conditions at the mRNA (Fig 1F) and protein (Fig 1G) level in the A549 cell line. This was confirmed in the bronchial epithelial cell line BEAS-2B (Fig S2A). Interestingly, the dominant GFRα receptor in epithelial cells is GFRα1 (Figs 1F and S2B). Enhanced production of NRTN was observed in A549 cells after stimulation with polyI:C and LPS (Fig 1G). Furthermore, increased NRTN was detected in the BAL fluid of influenza infected mice at time points associated with peak inflammation (Fig S2C). This data further supports a role for the GFRα2–RET–NRTN complex in host defence by lung macrophages. Moreover, GFRα2 expression by monocytes and macrophages suggests that this is not tissue specific or associated with a particular monocyte/macrophage lineage.

### Macrophage RET expression is indirectly induced by TLR agonists via type I IFNs

The TLR3 receptor agonist polyI:C triggers a downstream signalling cascade involving TANK-binding kinase 1 (TBK1) that induces the production of type I IFNs (Clark et al, 2011). In human MDMs, polyI:C enhanced IFNα and IFNβ mRNA (Fig S3A and B, respectively). However, in human lung macrophages, polyI:C significantly up-regulated the protein release (Fig 2A) and mRNA of IFNβ (Fig 2B), but not IFNα (Fig S3C). Blockade of TBK1 activation with BX795 abolished polyI:C-induced up-regulation of IFNβ (Fig 2A and B) and RET mRNA by human lung macrophages (Fig 2C). To determine whether RET is induced directly or indirectly by TLR3 activation, human lung macrophages (Fig 2D), human peripheral blood monocytes (Fig 2E) and human MDMs (Fig 2F) were stimulated with IFNs (IFNα, IFNβ, IFNγ, and IFNλ). RET expression was up-regulated by IFNβ in all three cell types at the mRNA level (Fig 2D–F). Furthermore, a slight increase in GFRα2 expression was observed in human lung macrophages by IFNγ (Fig S3D). The mRNA for the most common RET isoforms, RET 9 and RET 51, were both up-regulated by IFNβ, and to a lesser extent IFNα stimulation, in human lung macrophages (Fig S3E). This RET up-regulation was confirmed at the protein level in human MDMs using different concentrations of IFNβ (Fig 2G and H). GFRα2 protein, however, remained unchanged (Fig 2I). Although THP-1 monocytes and PMA-treated THP-1 macrophages expressed RET constitutively (Figs S3F and 1D, respectively),

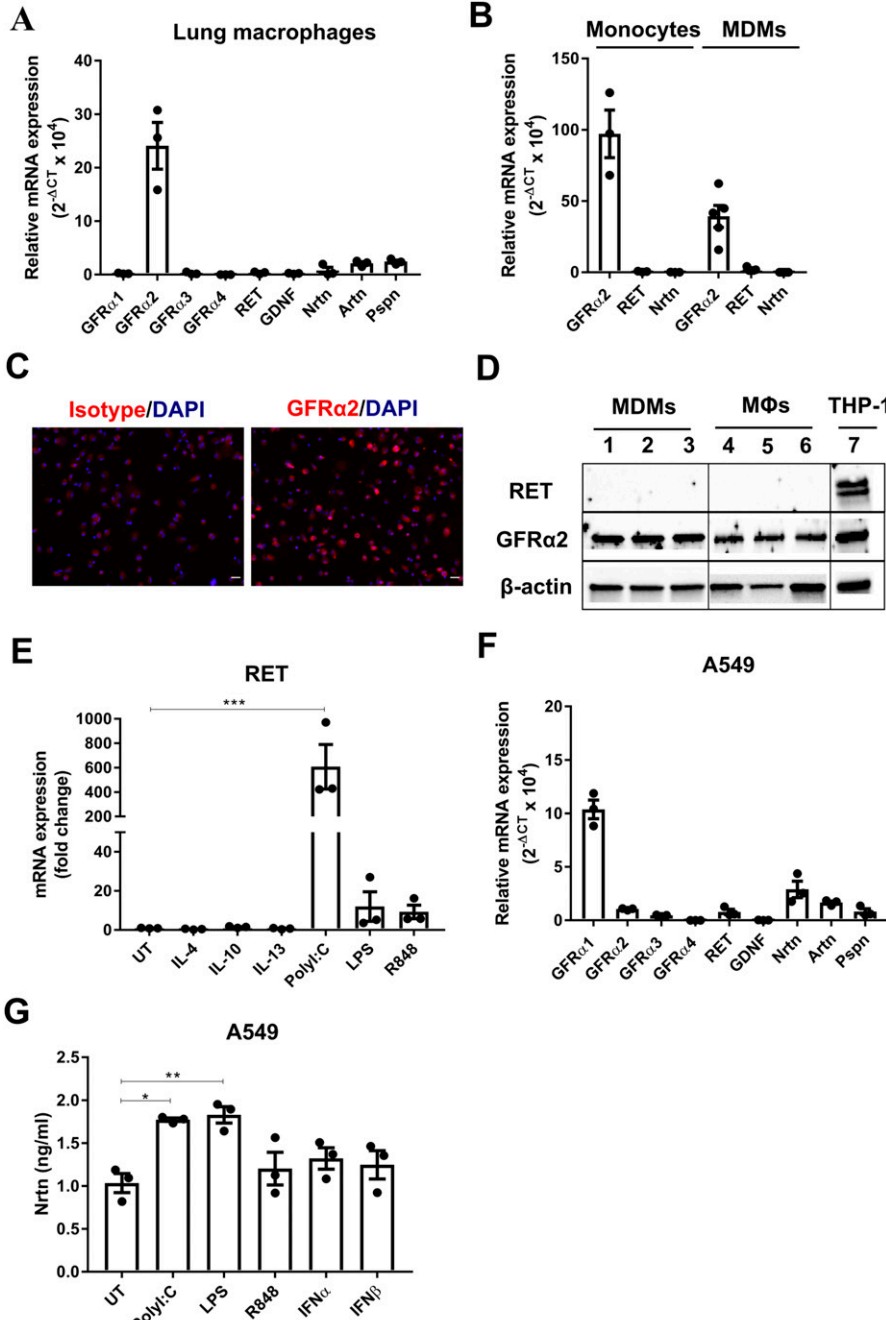

Figure 1. Human lung macrophages express GFRα2 and induce RET expression after TLR activation, whereas lung epithelial cells express the ligand NRTN.

**(A)** Relative mRNA expression ($2^{-\Delta CT} \times 10^4$) of GDNF family receptors and ligands in human lung macrophages (n = 3). **(B)** Relative mRNA expression of GFRα2, RET, and NRTN in peripheral blood monocytes (n = 3) or MDMs (n = 6) from non-matched donors. **(C)** Representative images of GFRα2 expression in human lung macrophages by immunofluorescence (GFRα2 or isotype control [red] DAPI [blue]). Scale bar = 20 µM. **(D)** Representative Western blot of GFRα2 and RET protein expression in MDMs (lanes 1–3) (n = 3), human lung macrophages (MΦs) (lanes 4–6) (n = 3) and PMA-treated THP-1 macrophage positive control (lane 7); β-actin housekeeping control. **(E)** mRNA expression of RET in MDMs after stimulation with IL-4, IL-10, IL-13 (all 20 ng/ml), polyI:C (10 µg/ml), LPS, or R848 (both 100 ng/ml) for 24 h (n = 3) expressed as a fold change over the average expression of untreated (UT) cells. **(F)** Relative mRNA expression ($2^{-\Delta CT} \times 10^4$) of GDNF family receptors and ligands in A549 cells (n = 3). **(G)** ELISA analysis of NRTN in A549 cells stimulated with polyI:C (10 µg/ml), LPS, R848 (both 100 ng/ml), IFNα or IFNβ (both 20 ng/ml) for 24 h (n = 3). Data information: Data are shown as mean ± SEM from three to six independent experiments. **(E, G)** *$P < 0.05$, **$P < 0.01$, ***$P < 0.001$; one-way ANOVA with Tukey's post hoc test with multiple comparisons (E, G). Source data are available for this figure.

IFNβ further enhanced RET protein in PMA-treated THP-1 macrophages (Fig 2J).

### NRTN up-regulates MMP2 production by human lung macrophages

To determine the function of NRTN on human lung macrophages, we ran a qPCR array analysis for genes related to macrophage function. In the case for macrophages from this donor, NRTN alone had mild effects on macrophages, including the up-regulation of

neuropeptide Y receptor type 2 (NPY2R) but down-regulation of NPY1R (Fig S4A). This mild effect of NRTN likely reflects that RET expression has not been induced and so we next examined the effect of NRTN on IFNβ-stimulated human lung macrophages. IFNβ alone increased mRNA for the granulocyte chemoattractant, CXCL8 and the lymphocyte chemoattractant, CXCL9 (Fig S4B). However, IFNβ stimulation of human lung macrophages with the addition of NRTN enhanced MMP2 mRNA and down-regulated IL-6, IFNγ, MAPK, MMP9, and IL-12A compared with IFNβ alone (Fig S4C). Gain-of-function mutations in RET lead to the development of specific

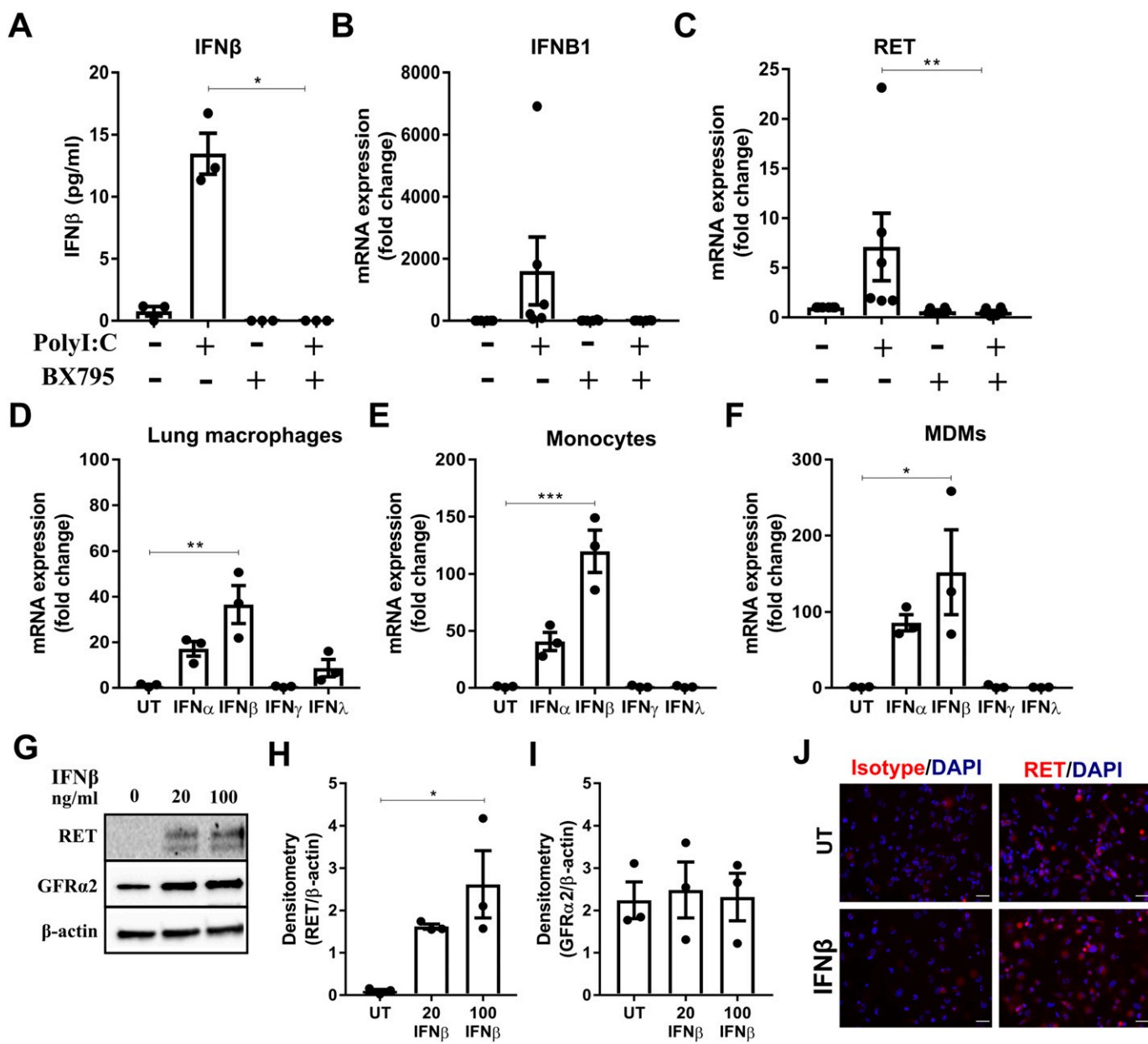

**Figure 2. Macrophage RET expression is indirectly stimulated by TLR agonists via type I IFNs.**
**(A, B, C)** ELISA analysis of IFNβ release (n = 3) (A) and mRNA expression of IFNβ (B) or RET (C) (n = 6) in human lung macrophages stimulated with polyI:C (10 µg/ml) ± BX795 (5 µg/ml) for 24 h. **(D, E, F)** RET mRNA expression in human lung macrophages (D), human peripheral blood monocytes (E), and human MDMs (F) stimulated with IFNα, IFNβ, IFNγ, or IFNλ (all at 20 ng/ml) for 24 h (n = 3). **(G, H, I)** Representative Western blot (untreated [lane 1], 20 ng/ml IFNβ [lane 2], 100 ng/ml IFNβ [lane 3]) (G) and densitometry analysis of RET (H) and GFRα2 (I) protein expression relative to β-actin in human MDMs stimulated with 20 ng/ml or 100 ng/ml of IFNβ for 24 h (n = 3). **(J)** Representative images of RET expression in untreated or IFNβ-treated PMA-treated THP-1 macrophages (top panels; untreated [UT], bottom panels; IFNβ-treated [20 ng/ml], RET [red] DAPI [blue]). Scale bar = 40 µM. Data information: Data are shown as mean ± SEM from three to six independent experiments. **(A, C, D, E, F, H)** *P < 0.05, **P < 0.01, ***P < 0.001; one-way ANOVA with Tukey's post hoc test with multiple comparisons (A, D, E, F, H); Kruskal–Wallis test with Dunn's multiple comparisons test (C). Source data are available for this figure.

human cancers and have been associated with the regulation of MMPs, including MMP2 and MMP9, which are thought to drive metastasis (Kato et al, 1998; Asai et al, 1999). To validate the effect of NRTN on MMP2 production, human lung macrophages were stimulated with polyI:C or IFNβ, with or without NRTN. PolyI:C-stimulated human lung macrophages, with the addition of NRTN, significantly

enhanced the production of MMP2 (~140 kD, homodimer) (Fig 3A and B). In contrast to the changes in MMP2 at the mRNA level, NRTN did not enhance MMP2 production in IFNβ-stimulated human lung macrophages (Fig 3B). The expression of MMP9 was also analysed as this MMP belongs to the same group as MMP2, the gelatinases; however, MMP9 (~92 kD) was unchanged by NRTN stimulation (Fig

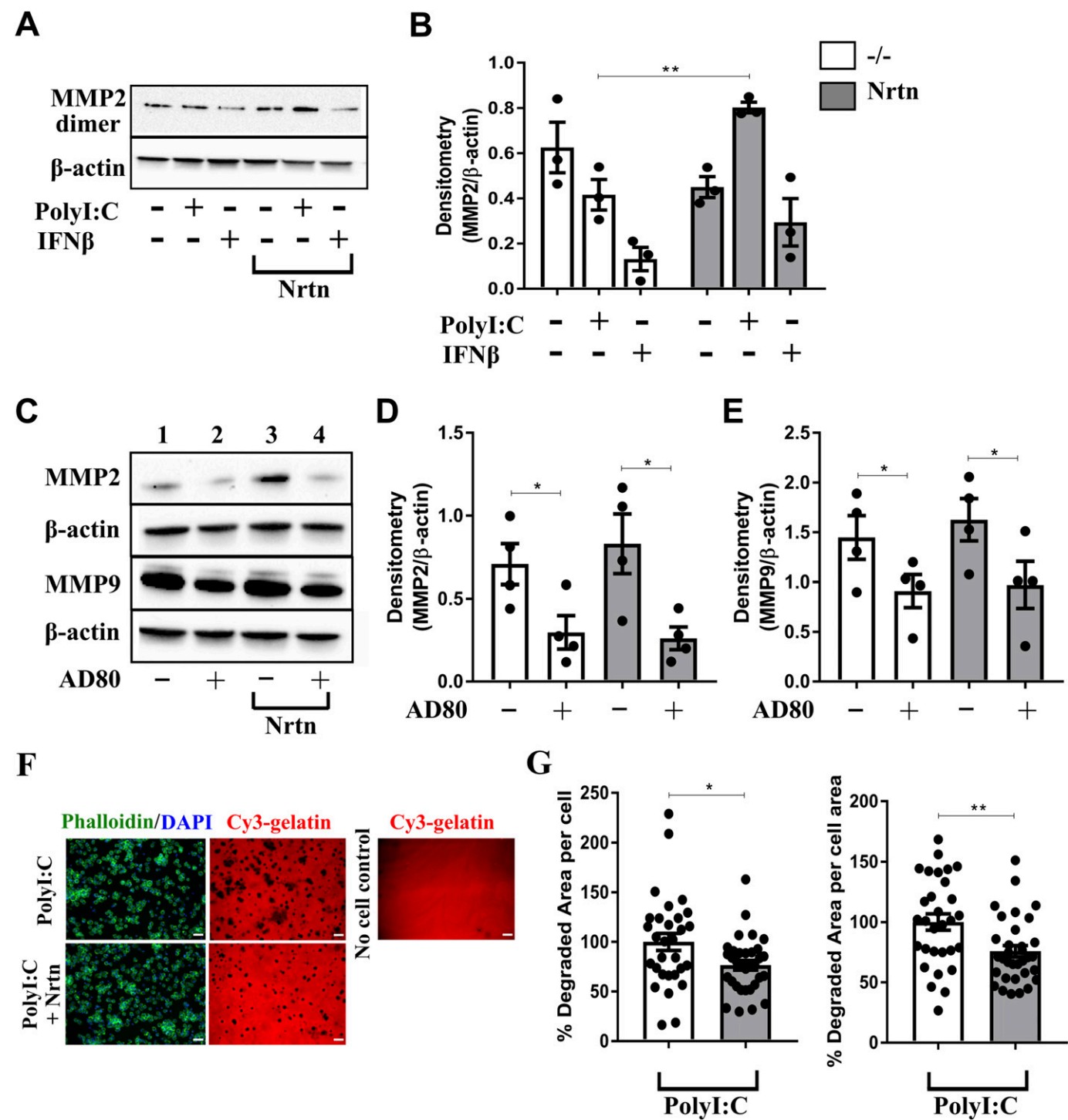

**Figure 3. NRTN promotes MMP2 and MMP9 production from macrophages.**
**(A, B)** Representative Western blot (A) and densitometry analysis (B) of MMP2 in human lung macrophages treated with polyI:C (10 µg/ml) or IFNβ (20 ng/ml) ± NRTN (100 ng/ml) for 24 h (n = 3). **(C, D, E)** Representative Western blot (C) and densitometry analysis of MMP2 (D) or MMP9 (E) protein expression in PMA-treated THP-1 macrophages stimulated with AD80 (0.5 µM) ± NRTN (100 ng/ml) for 24 h (n = 4). Densitometry analysis of MMP2 and MMP9 is expressed relative to β-actin. **(F, G)** Representative image (F) and quantification per cell and per cell area (G) of gelatin degradation by human lung macrophages stimulated with polyI:C (10 µg/ml) ± NRTN (100 ng/ml) for 24 h (n = 3). Each dot on the graph represents a field of view, minimum seven fields of view per donor. Scale bar = 50 µM. Data information: Data are shown as mean ± SEM from three or four independent experiments. **(B, D, E, G)** *P < 0.05, **P < 0.01, ***P < 0.001; Student's paired two-tailed t test (B, D, E, G).
Source data are available for this figure.

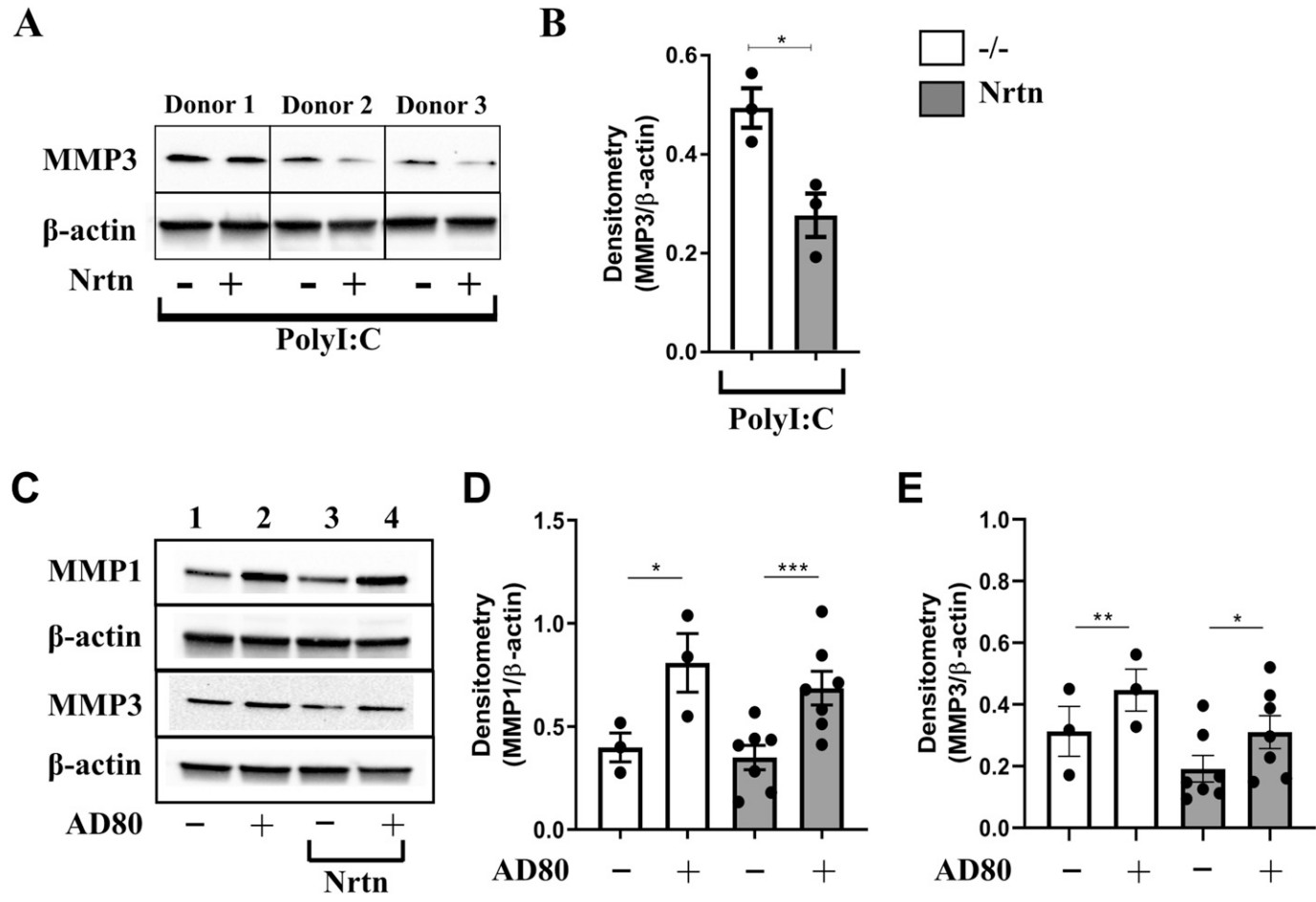

**Figure 4.   NRTN reduces MMP1 and MMP3 production from macrophages.**
**(A, B)** Representative Western blot (A) and densitometry analysis of MMP3 (B) in human lung macrophages stimulated with polyI:C (10 μg/ml) ± NRTN (100 ng/ml) for 24 h (n = 3). **(C, D, E)** Representative Western blot (C) and densitometry analysis of MMP1 (D) or MMP3 (E) protein expression in PMA-treated THP-1 macrophages stimulated with AD80 (0.5 μM) ± NRTN (100 ng/ml) for 24 h (n = 3–7). Densitometry analysis of MMP1 and MMP3 is expressed relative to β-actin. Data information: Data are shown as mean ± SEM from three independent experiments. **(B, D, E)** *P < 0.05; Student's paired two-tailed t test (B, D, E).
Source data are available for this figure.

S4D and E). Interestingly, IFNβ significantly reduced MMP9 protein release from lung macrophages (Fig S4F), although this was not observed at the mRNA level (Fig S4B).

To determine whether inhibition of RET leads to the down-regulation of MMP2 production in macrophages, the RET inhibitor, AD80, was used in PMA-treated THP-1 macrophages. A reduction in MMP2 (~72 kD) and MMP9 (~92 kD) production following treatment with AD80 was observed in both the presence and absence of NRTN (Fig 3C–E). To investigate whether the changes in MMP2 production induced by NRTN in polyI:C-stimulated macrophages had a functional effect on MMP2 activity we performed a gelatin degradation assay. A reduction in MMP activity from polyI:C and NRTN stimulated human lung macrophages was observed as measured by their reduced ability to degrade gelatin when analysed per number of cells or by cell area, compared with polyI:C alone (Fig 3F and G). This reduction in gelatin degradation did not correlate with the increase in MMP2 production observed and suggests that NRTN via RET ligation may either negatively affect MMP2 activity or regulate the production of other MMPs that can degrade gelatin.

**NRTN down-regulates MMP1 and MMP3 production by macrophages**

To determine whether NRTN can regulate the production of other types of MMPs by human lung macrophages, the production of the interstitial collagenase, MMP1, and the stromelysin, MMP3, was analysed. The production of MMP1 and MMP3 has been previously linked to RET activation (Mesa et al, 2006; Kang et al, 2009) and gelatin is a known substrate for both of these MMPs. PolyI:C-stimulated human lung macrophages, with the addition of NRTN, down-regulated the production of MMP3 (~20 kD form) compared with polyI:C alone (Fig 4A and B). MMP1 was not expressed by human lung macrophages (data not shown). Conversely RET inhibition by AD80, in the presence or absence of NRTN, in PMA-treated THP-1 macrophages increased the expression of MMP1 (~53 kD) and MMP3 (~54 kD) (Fig 4C–E). Overall, these data show that RET activation via Gfrα2 and NRTN decreases the production of MMP3 and promotes MMP2 in primary human lung macrophages but MMP activity may also be affected. Our data suggest that RET could switch the

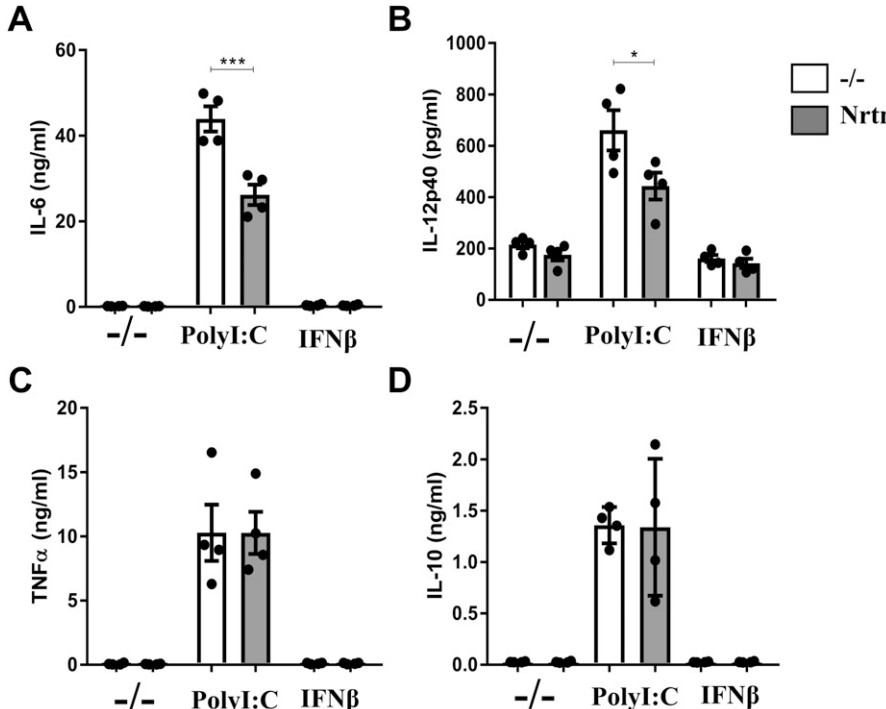

**Figure 5. NRTN dampens the release of pro-inflammatory cytokines from human lung macrophages.**
**(A, B, C, D)** ELISA analysis of IL-6 (A), IL-12p40 (B), TNFα (C) and IL-10 (D) release from human lung macrophages stimulated with polyI:C (10 μg/ml) or IFNβ (20 ng/ml) ± NRTN (100 ng/ml) for 24 h (n = 4). Data information: Data are shown as mean ± SEM from four independent experiments. **(A, B)** *P < 0.05, ***P < 0.001; one-way ANOVA with Tukey's post hoc test with multiple comparisons (A, B).
Source data are available for this figure.

production of MMPs produced by macrophages depending on the demands of the tissue at that time.

### NRTN dampens the release of pro-inflammatory cytokines from human lung macrophages

The GDNF family is a member of the wider TGFβ superfamily and as such have been hypothesised to have anti-inflammatory properties. In our system, NRTN significantly decreased the release of IL-6 (Fig 5A) and IL-12p40 (Fig 5B), but not TNFα (Fig 5C) or IL-10 (Fig 5D), from polyI:C-stimulated human lung macrophages. This suggests that NRTN may prevent excessive inflammation and limit further tissue damage during a viral infection. The data also corroborate an anti-inflammatory role for the GDNF family.

### GDNF family expression is enhanced in macrophages from the lungs of chronic smokers and tumour tissue

MMPs are implicated in a range of chronic airway diseases, such as asthma, chronic obstructive pulmonary disease (COPD) (Hendrix & Kheradmand, 2017), and cancer metastases (Kessenbrock et al, 2010). GFRα2, RET, and MMP2 mRNA expression was analysed in lung macrophages from patients with COPD compared with patients with no underlying respiratory disease (Fig 6A–C). No significant differences could be identified. However, if patients were categorised based on their smoking status, a significant increase in GFRα2 and MMP2 mRNA was detected in lung macrophages from current smokers compared with non-smokers, regardless of disease status (Fig 6D–F). A recent genome-wide association study study found that RET expression is linked to the number of cigarettes smoked per day (Matoba et al, 2019); however, we did not see enhanced RET mRNA expression in lung

macrophages from smokers compared with non-smokers in our smaller study group (Fig 6E). GFRα2 and RET mRNA was also compared between macrophages isolated from lung tumours and surrounding healthy margin tissue. Although no difference was detected in GFRα2 mRNA (Fig 6G), RET mRNA was increased in tumour compared with margin macrophages (Fig 6H). These data suggest that the GDNF family may be important factors to investigate further in the context of smoking/COPD and lung cancer, with regard to their effect on MMP production.

## Discussion

Our data demonstrate that NRTN drives a functional switch in human lung-resident macrophages that inhibits inflammatory cytokine production and controls the expression of MMPs (Fig 7). That Gfrα2 was expressed on monocytes and macrophages irrespective of their stage of differentiation, tissue location, or subtype, which suggests a fundamental role in macrophage biology. Regulation of MMP production via this pathway relies on viral triggers to promote NRTN production from epithelial cells and type I IFN to up-regulate the signalling adapter RET in macrophages. Collectively, the data imply an important role for this multi-component receptor in airway host defence.

Accumulating evidence describes a role for the GDNF family in shaping an antimicrobial immune response. Intestinal ILC3s are critical regulators of host defence (Artis & Spits, 2015) that constitutively express RET and respond to neurotrophic factors secreted by glial cells (Ibiza et al, 2016). In this case, the outcome is IL-22 release that in turn initiates repair genes in epithelial cells and promotes host defence. The induction of RET by type I IFNs that we

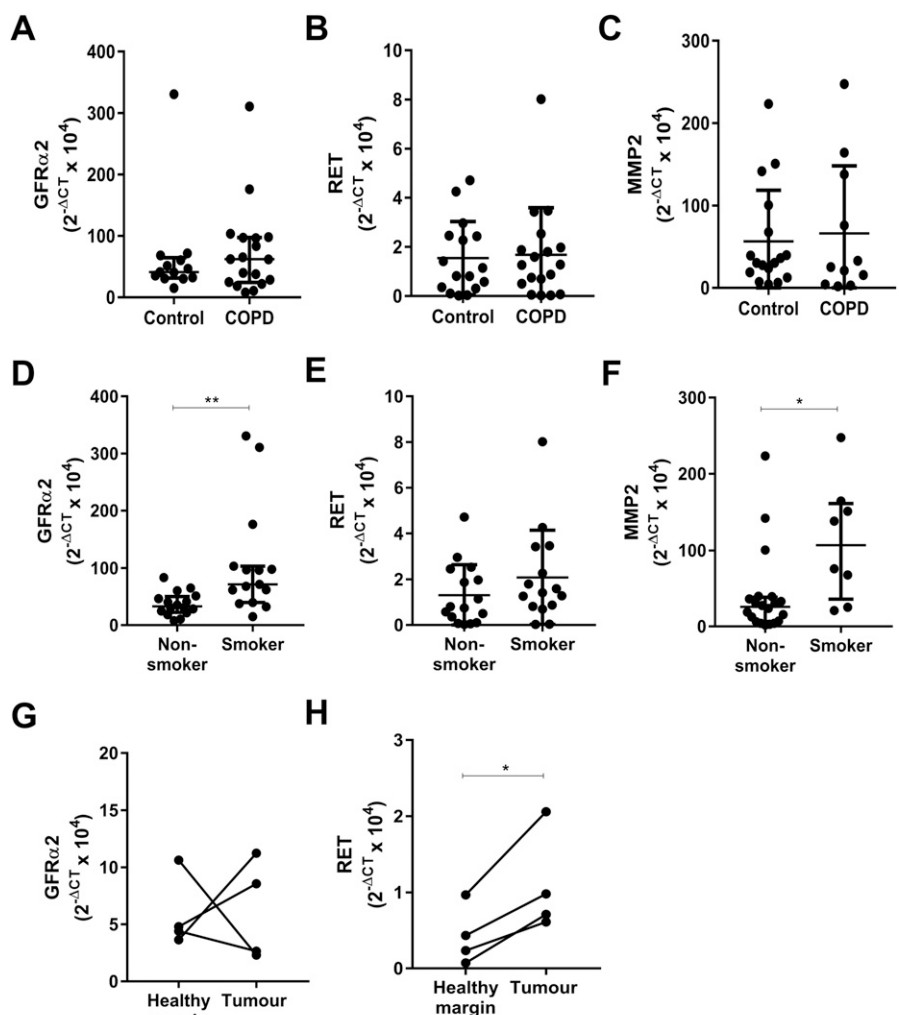

**Figure 6. GDNF family expression is enhanced in macrophages from the lungs of chronic smokers and lung tumour tissue.**
**(A, B, C)** Relative GFRα2 (A), RET (B), and MMP2 (C) mRNA expression ($2^{-\Delta CT} \times 10^4$) in macrophages isolated from the lungs of patients categorised as "controls" (n = 16) or chronic obstructive pulmonary disease (GFRα2 and RET, n = 18; MMP2, n = 11). **(D, E, F)** Relative GFRα2 (D), RET (E), and MMP2 (F) mRNA expression in macrophages isolated from the lungs of patients categorised as non-smokers (n = 18) or smokers (GFRα2 and RET, n = 16; MMP2, n = 8). **(G, H)** Relative mRNA expression ($2^{-\Delta CT} \times 10^4$) of GFRα2 (G) and RET (H) in macrophages isolated from lung tumour tissue and matching healthy margin tissue (n = 4). **(A, B, C, D, E, F)** Data information: Data are shown as median ± interquartile range (A, B, C, D, E, F). **(D, F, H)** *$P < 0.05$, **$P < 0.01$; Mann–Whitney U test (D, F), Student's paired two-tailed $t$ test (H).
Source data are available for this figure.

observe in macrophages implicates the GDNF family in mediating host defence in the respiratory tract. We show that RET activation in human lung macrophages by NRTN alters the production of MMPs by enhancing MMP2 and reducing MMP3 production. We also observe that RET inhibition in THP-1 macrophages decreases the production of MMP2 and MMP9 and enhances the production of MMP1 and MMP3. RET activation may therefore provide a critical hub for the control of macrophage MMP production. The MMPs are a family of zinc-dependent endopeptidases and have various important functions in infection, disease, repair and lung development (Beeh et al, 2003; Belvisi & Bottomley, 2003; Hendrix & Kheradmand, 2017). MMPs are expressed at relatively low levels at homeostasis but are up-regulated during inflammation and their production must be tightly controlled (Loffek et al, 2011). One important role of MMPs is the degradation of the ECM and interestingly in NRTN knockout mice, enhanced collagen deposition is evident around the airways, although the mechanism is unknown (Mauffray et al, 2015). Our data suggest that this may be due to the effect of NRTN on lung macrophage MMP production.

MMPs have fundamental roles in immune cell trafficking to sites of infection and this is achieved through cleaving transmembrane-

and matrix-embedded proteins, including chemokines, and regulating the activity of cytokines (Sternlicht & Werb, 2001; Overall, 2002). The MMPs we studied have been implicated in the activation of the pro-inflammatory cytokines TNFα and IL-β and the anti-inflammatory TGFβ (Gearing et al, 1994; Ito et al, 1996; Schönbeck et al, 1998; Dallas et al, 2002; Karsdal et al, 2002; Maeda et al, 2002). Pro-TNFα is cleaved to its active form by MMP1, MMP2, MMP3 and MMP 9 (Gearing et al, 1994), whereas MMP2, MMP3, and MMP9 cleave latent TGFβ-binding protein-1 to release TGFβ (Dallas et al, 2002; Maeda et al, 2002) and may also directly activate the latent form of TGFβ (Karsdal et al, 2002). MMP2, MMP3, and MMP 9 can both positively and, with the addition of MMP1, negatively regulate the activation of IL-1β (Ito et al, 1996; Schönbeck et al, 1998). The monocyte chemoattractant proteins, CCL2 (MCP-1), CCL7 (MCP-3), CCL8 (MCP-2), and CCL13 (MCP-4), are all regulated by MMPs. MMP2 cleaves CCL7 to an inactive form (McQuibban et al, 2000), and MMP1 and MMP3 cleave the active forms of CCL2, CCL8, and CCL13 thereby reducing their chemoattractant activity (McQuibban et al, 2002). Our data add another level of complexity to the regulation of cytokines and chemokines by MMPs. We have shown that macrophage RET expression is tightly controlled and that activation via neurturin

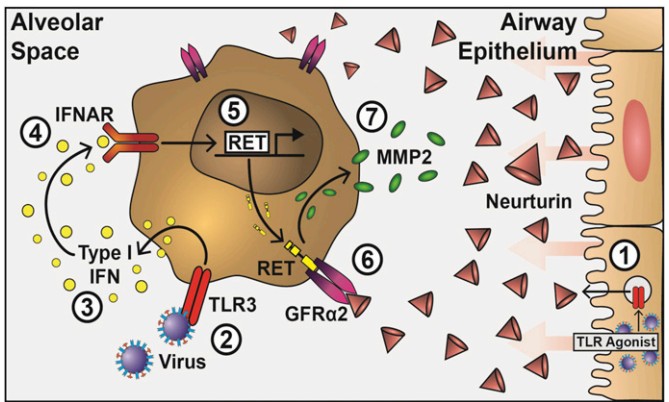

**Figure 7. A novel role for the GDNF family on macrophages in respiratory viral infection.**
TLR activation of airway epithelial cells enhances the production of NRTN, which binds to constitutively expressed GFRα2 on the surface of lung macrophages (1). Viral TLR activation on lung macrophages (2) triggers the release of IFNβ (3), which acts in an autocrine manner and binds to the IFNα/β receptor, IFNAR (4), to stimulate the production of RET at the mRNA and protein level (5). RET translocates to the cytoplasm and binds to GFRα2 (6), stimulating the production of MMP-2 (7) and dampening the pro-inflammatory cytokine response.

enhances MMP2 production and decreases MMP3 production in TLR3-activated lung macrophages. This may promote the release of active TGFβ and reduce the chemotactic potential of CCL7 to attract monocytes to the site of infection. However, with the crossover in cytokine and chemokine regulation by MMP2 and MMP3, it remains unclear how the effect of NRTN on the production of MMP2 and MMP3 by macrophages influences the inflammatory response to an infection in vivo.

Our knowledge of the roles of individual MMPs is still limited; however, several studies have implicated specific MMPs in respiratory disease. MMP2 has beneficial effects in a mouse model of asthma, where this MMP facilitates the resolution of allergic inflammation by allowing cells to egress to the lung (Corry et al, 2004). On the other hand, in the absence of MMP2 and/or MMP9, specific chemotactic gradients are disrupted and cause retention of cells in the lung parenchyma and inhibition of their exit via the airways (Haas & Madri, 1999; Corry et al, 2002; Li et al, 2002; McGuire et al, 2003; Parks, 2003). Conversely, MMP3 has a detrimental role in mouse models of acute lung injury and pulmonary fibrosis, with MMP3-deficient mice exhibiting less severe lung injury and protection from bleomycin-induced fibrosis (Warner et al, 2001; Yamashita et al, 2011). Although further studies are required to elucidate the roles of MMPs in respiratory viral infections, it has been shown in a mouse model of severe influenza infection (IAV) that MMP9$^{-/-}$ mice exhibited increased survival and were protected from IAV-induced lung injury (Rojas-Quintero et al, 2018). This suggests that a reduction in MMP9 expression would be beneficial in host defence against respiratory viral infections. Interestingly, induction of MMP9 expression in macrophages of IAV-infected mice has been shown to be via the TLR7 pathway (Lee et al, 2017). This may explain why we did not observe an effect of NRTN on MMP9 expression in TLR3-activated macrophages. Although our this study is limited to in vitro experiments, our data suggest that in vivo the

GDNF family, through the regulation of MMPs, could affect the activity of cytokines and chemokines, influencing the cells that infiltrate the airways in response to a viral infection.

When investigating MMP expression at the mRNA and protein levels in our system, we noted some discrepancies, which highlights the need to validate expression at both molecular levels. We observed enhanced MMP2 at the mRNA level in IFNβ-stimulated macrophages by NRTN; however, NRTN with polyI:C stimulation, but not IFNβ stimulation, enhanced MMP2 protein expression. It is not clear why we see this difference but suggests that other factors produced by TLR3 stimulation may be required for the alteration in MMP expression at the protein level induced by NRTN, which are not produced by IFNβ stimulation alone. In addition, MMP9 release from lung macrophages was reduced by IFNβ at the protein level; however, no differences were observed at the mRNA level. The disparity in mRNA and protein levels observed may be due to the different time points analysed or suggests that IFNβ has post-translational effects on MMP9.

The GDNF family are members of the wider TGFβ superfamily (Airaksinen & Saarma, 2002) and as such studies have investigated their anti-inflammatory functions. GFLs modulate T helper 2 (Th2) cells by decreasing IL-10 production in a RET-dependent manner (Almeida et al, 2014), NRTN knockout mice develop exaggerated inflammation to allergens (Mauffray et al, 2015) and addition of GDNF or NRTN to activated PBMCs reduces TNF protein (Vargas-Leal et al, 2005). In corroboration, our data demonstrates that in the setting of a viral infection, NRTN dampens the release of pro-inflammatory cytokines from human lung-resident macrophages. Overall, the outcome of GFRα2-NRTN-RET signalling in immune cells appears to limit excessive inflammation and drive inflammatory resolution.

Our data suggest that the GDNF family may be important factors to investigate in airway disease. We observed enhanced GFRα2 and MMP2 expression in lung macrophages from current smokers. Whether this impacts on the development of COPD, or exacerbations of the disease, remains to be elucidated. The glial cell line derived neurotrophic factor family were first characterised as neuronal growth factors involved in the development of the nervous system, in particular the enteric nervous system (Airaksinen & Saarma, 2002). RET activation has not been directly linked to MMPs in nervous system formation; however, MMP2 is involved in the migration and network formation of enteric neural crest-derived cells (Anderson, 2010). The production of matrix proteins via RET activation has been better characterised in the context of cancer (Mulligan, 2018). Gain-of-function mutations in RET are associated with familial neuroendocrine tumours and medullary thyroid cancers (Jhiang, 2000). Deletion of RET in thyroid carcinomas has been shown to reduce the production of MMP2 and MMP9 (Lian et al, 2017). However, RET expression by immune cells in cancer has, to the best of our knowledge, not been previously assessed. MMPs are known to alter the microenvironment to aid cancer progression and metastases (Kessenbrock et al, 2010). Our data suggest that RET could be up-regulated in tumour macrophages, thereby limiting inflammation and enhancing MMP2 production within the tumour microenvironment. RET inhibition may therefore be an attractive therapeutic target in cancer as an immune modulator irrespective of the presence of RET-activating mutations.

All epithelial surfaces have a dense network of nociceptors that enable them to sense noxious substances, cytokines, and pathogens (via pathogen-associated molecular patterns). In turn, neuronal factors released from peripheral neurons influence local cell recruitment, their activation and ultimately the profile of the immune response (for a review, see Talbot et al [2016]). This integrated system provides a rapid response to threat, which if reliant on the immune system alone, could take days to mobilize. Although NRTN redirected macrophage activity without the presence of neurons, in vivo we have not ruled out that epithelial-derived NRTN may influence local sensory neurons or that NRTN could be produced by other cell types, such as glial cells that have now been described in the lung (Suarez-Mier & Buckwalter, 2015). The data suggest that epithelial cells may be able to communicate to neurons the type of threat involved and that neurons in turn may direct an appropriate response.

The alteration in cytokine and matrix protein production by the GFRα2-RET-NRTN complex that we describe provides many avenues for therapeutic modulation if the goal is to modulate specific MMP activity or inflammatory cytokine responses from macrophages. This may have the advantage of releasing the inflammatory potential of airway macrophages so that they are quicker at quelling an infectious agent. Further studies investigating the GDNF family in vivo may also elucidate the roles of specific MMPs and determine whether this is actually detrimental, or beneficial for resolution of inflammation.

# Materials and Methods

### Study participants

Regions of healthy marginal tissue (>6 cm from the cancer) were identified and dissected by a histopathologist under auspices of the Manchester Allergy, Respiratory and Thoracic Surgery (Man-ARTS) Biobank at the University Hospital of South Manchester. Ethical approval was granted by the National Research Ethics Service Committee (ref; 15/NW/0409). Participants with a forced expiratory volume in 1 s/forced vital capacity ($FEV_1$/FVC) of ≥0.70 with no other underlying respiratory disease were categorised as "healthy." Participants with COPD were defined by physician diagnosis and exhibited an $FEV_1$/FVC < 0.70. Participants were also categorised based on smoking status and demographics as detailed in Table S1. Human lung macrophages from "healthy" non-smokers were used for experiments unless otherwise stated. Lung tumour samples were collected from the Manchester Cancer Research Centre Biobank under the Manchester Cancer Research Centre Biobank Research Tissue Bank Ethics (ref; 18/NW/0092). All patients provided written consent for participation in the studies.

### Isolation of airway macrophages from human lung resection samples

Sterile PBS was flushed through resected lung tissue using a 26 G needle and syringe until the liquid ran clear as described previously (Dewhurst et al, 2017). Cells were layered onto Ficoll-paque (GE Healthcare) the leukocyte cell layer collected, washed in sterile PBS

and counted using trypan blue and a haemocytometer. Cells were plated in complete RPMI media (RPMI 1640, 10% FCS, 100 U/ml penicillin, and 100 µg/ml streptomycin [all from Sigma-Aldrich]) and incubated for 1 h at 37°C to allow macrophages to adhere. For RNA and protein extraction cells were plated at $5 × 10^5$ cells/ml in a 12- or 6-well plate, respectively. For imaging, cells were plated at $1.5 × 10^5$ cells/ml on coverslips in a 24-well plate. Cells were washed three times with sterile PBS, media replaced and stimulated with polyI:C (10 µg/ml) (InvivoGen), IFNα, IFNβ, IFNγ or IFNλ (all used at 20 ng/ml), or NRTN (100 ng/ml) (all from Peprotech) and incubated for 24 h at 37°C before further processing.

### Isolation of macrophages from tumour tissue and matching healthy margin tissue

Lung and tumour tissue was finely chopped and enzymatically digested with Liberase TL (0.125 mg/ml; Sigma-Aldrich) and DNase (150 U/ml; Sigma-Aldrich) in HBSS (Sigma-Aldrich) in a shaking incubator at 37°C for 30 min. Tissue was passed through a 70 µM filter and the cell suspension centrifuged at 400$g$ for 5 min. Cells were resuspended in sterile distilled water for 30 s to lyse red blood cells. Cells were centrifuged and washed with sterile PBS and counted using trypan blue and a haemocytometer. Cells were plated at $1.25 × 10^5$ cells/ml in a 24-well plate and incubated for 1 h at 37°C to allow macrophages to adhere. Cells were washed three times with sterile PBS and lysed in RLT buffer (QIAGEN) for RNA extraction.

### Isolation and culture of MDMs

Leukocyte apheresis cones from healthy donors were provided by the national blood transfusion service, layered onto Ficoll-paque (GE Healthcare) and leukocyte cell layer collected. Cells were washed in sterile PBS and counted using trypan blue and a haemocytometer. PBMCs were incubated with anti-human CD14 magnetic beads (Miltenyi Biotec) and sterile MACS buffer (sterile PBS, 0.5% BSA, 2 mM EDTA) and passed through an LS MACS separation column in a MidiMACS separator according to the manufacturer's instructions (Miltenyi Biotec) to obtain purified CD14$^+$ monocytes. CD14$^+$ monocytes were plated at $0.75 × 10^6$ cells/ml in a 12-well plate for RNA isolation or a six-well plate for protein extraction and cultured in complete RPMI media with M-CSF (50 ng/ml) (Peprotech). Half of the media was replaced after 3 d without the addition of M-CSF. After 6 d, media was replaced and differentiated macrophages stimulated with LPS (100 ng/ml), polyI:C (10 µg/ml), R848 (100 ng/ml) (all from InvivoGen), IFNγ, IL-4, IL-10, IL-13, IFNα, IFNβ, or IFNλ (all 20 ng/ml and from Peprotech) and incubated for 24 h at 37°C before further processing.

### Mouse model of influenza virus infection

8- to 12-wk-old C57BL/6 mice (Harlan Olac Ltd) were kept in specific pathogen-free conditions at Bio Safety Level 2 with a controlled temperature (21°C ± 1°C), humidity (55% ± 10%), and a 12-h light/dark cycle, with food and water ad libitum. All animal procedures were approved by the Home Office UK and by the University of Manchester Animal Welfare and Ethical Review Body, and conformed to the requirements of the UK Animals (Scientific Procedures) Act, 1986. Mice were lightly anaesthetised with isoflurane and infected intranasally with 7.5 plaque forming units of influenza A virus, Puerto Rico/8/

34(PR8), H1N1. Mice were euthanized by intraperitoneal injection of 3 mg pentobarbitone at specific time points post-infection. A 21 G needle attached to a 0.2-mm catheter tube was inserted into the mouse trachea and 3 × 1 ml of HBSS containing 0.05 M EDTA (Sigma-Aldrich) was flushed into the lungs to harvest BAL fluid and macrophages as previously described (Snelgrove et al, 2008).

## Isolation and culture of BMDMs

Hind limbs were removed and the bone stripped of skin and muscle. The tips of the femur and tibia were cut and bone marrow flushed out with sterile PBS. Cells were centrifuged at 500$g$ for 5 min, supernatant discarded and the pellet resuspended in 3 ml of red blood cell lysis buffer (Sigma-Aldrich) for 3 min. After centrifugation, cells were washed in sterile PBS and resuspended in complete RPMI media with 20% FCS (Gibco) supplemented with GM-CSF (20 ng/ml) or M-CSF (20 ng/ml) (both Peprotech) and cultured for 3 d at 37°C. Media was replaced with complete RPMI media containing 10% FCS and cells further cultured for 4–7 d until confluent. Cells were lysed in RLT buffer and stored at −80°C until RNA extraction.

## Cell lines

THP-1 cells were cultured in complete RPMI media and plated at 5 × 10$^5$ cells/ml in a six-well plate or at 2 × 10$^5$ cells/ml on fibronectin coated coverslips in a 24-well plate for imaging. THP-1 cells were treated with 25 nM phorbol 12-myristate 13-acetate (PMA) (Sigma-Aldrich) to induce differentiation into macrophages. After 48 h, fresh complete RPMI medium was added to the cells and the plate incubated at 37°C overnight before stimulation. The medium was replaced and THP-1 cells were stimulated with IFN$\beta$ (20 ng/ml), NRTN (100 ng/ml) (both from Peprotech), and/or AD80 (Selleckchem) and incubated for 24 h at 37°C before further processing. A549 cells were plated at 0.75 × 10$^5$ cells/ml in a 12-well plate in complete DMEM medium and incubated at 37°C overnight before stimulation. A549 cells were stimulated with polyI:C (10 $\mu$g/ml), LPS (100 ng/ml), R848 (100 ng/ml) (all InvivoGen), or IFN$\alpha$ or IFN$\beta$ (both 20 ng/ml and from Peprotech) and incubated for 24 h at 37°C before further processing.

## RNA extraction and qPCR

RNA was isolated using the RNeasy micro kit according to the manufacturer's instructions (QIAGEN) and total RNA was quantified using a NanoDrop spectrophotometer (Thermo Fisher Scientific). Reverse transcription of equivalent amounts of RNA was carried out using the High-Capacity RNA-to-cDNA kit according to the manufacturer's instructions (Applied Biosystems). qPCR reactions were performed using TaqMan Fast Universal PCR master mix and predesigned Taqman expression assays (both from Life Technologies) or SybrGreen master mix (Applied Biosystems) on a Quantstudio 12k flex PCR system (Life Technologies). The RT$^2$ Profiler qPCR array for mouse neurotrophins and receptors was carried out according to the manufacturer's instructions. 1 $\mu$g of RNA from each sample was reverse transcribed using the RT2 First Strand Kit (QIAGEN) and qPCR reactions were performed using RT2 SybrGreen Rox qPCR

Master Mix. Data were analysed using the SABiosciences RT2 Profiler qPCR Array Data Analysis version 3.5. The GeneQuery Human Macrophage Cell Biology qPCR Array Kit was carried out according to the manufacturer's instructions. 250 ng cDNA was used in each well of the 96-well plate provided. Relative mRNA expression was calculated based on the ΔΔCT method (Livak & Schmittgen, 2001) using QuantStudio 12K Flex Software v1.1.1 (Life Technologies).

The average mRNA expressions of the housekeeping genes RN18s and Hprt were used to calculate relative mRNA levels in mouse BMDMs and mouse alveolar macrophages. The mRNA expression of RPLP0 was used to calculate relative mRNA levels in human monocytes, MDMs, BEAS-2B cells, A549 cells, THP-1 cells, and human lung, tumour, and sputum macrophages. RET9 was detected using the following primer pair: forward primer – 5′TCCCTTCCAC-ATGGATTG-3′; reverse primer – 5′-ATCACAGAGAGGAAGGATAGT-3′ and RET 51 was detected using the following primer pair: forward primer – 5′-CTCCCTTCCACATGGATTG-3′; reverse primer – 5′-TCAGCT-CTCGTGAGTGGT-3′. Gene expression was normalised to the housekeeping gene GAPDH using the following primer pair: forward Primer, 5′-GAAGGTGAAGGTCGGAGT-3′, reverse Primer, 5′-CATGGGTGGAATCA-TATTGGAA-3′.

## Protein extraction and Western blot analysis

Human lung macrophages, MDMs and THP-1 macrophages were lysed in RIPA buffer (50 mM Tris–HCL [pH 7.4], 1% NP-40, 0.25% Sodium Deoxycholate, 150 mM NaCl, and 1 mM EDTA) with protease inhibitors (Protease Inhibitor Cocktail; Sigma-Aldrich). Protein concentrations were determined using a BCA Protein Assay Kit (Thermo Fisher Scientific). Equivalent amounts of protein (50 $\mu$g) were added to 4× Laemmli sample buffer (2% SDS, 20% glycerol, 0.2% bromophenol blue, 1 M Tris–HCl [pH 6.8]). To detect RET in MDMs, 4% SDS in 1× laemmli sample buffer was used. Samples were loaded into 4–20% mini-Protean TGX precast gels (Bio-Rad) and resolved by electrophoresis and transferred to nitrocellulose membranes (Bio-Rad) using a semi-dry Trans-Blot turbo transfer system (Bio-Rad). The membrane was blocked with 5% milk (Sigma-Aldrich) and 0.1% Tween 20 (Sigma-Aldrich) in PBS for 1 h at room temperature and incubated overnight at 4°C with antibodies for GFR$\alpha$2 (1:500; R&D Systems), RET, MMP1, MMP2, MMP3, and MMP9 and pro-MMP9 (all used at 1:1,000; Cell Signalling Technology). Membranes were washed three times with 0.1% Tween 20 in PBS for 10 min and anti-rabbit HRP-conjugated secondary antibody (1:3,000; Dako) diluted in 5% milk and 0.1% Tween 20 in PBS was added and incubated for 1 h at room temperature on an orbital shaker. $\beta$-actin-peroxidase (1:10,000; Sigma-Aldrich) was used as the housekeeping control. The membranes were developed using a Clarity Western ECL Substrate (Bio-Rad) and visualised with a ChemiDoc MP Imaging System (Bio-Rad).

## Immunocytochemistry and immunofluorescence

Cells were fixed in 4% paraformaldehyde for 20 min, washed in PBS, and lung macrophages incubated in 10% donkey serum in PBS and THP-1 macrophages incubated in 5% goat serum and 0.3% Triton X-100 in PBS to for 1 h at room temperature to block non-specific binding. Blocking buffer was removed and wells incubated with

GFRα2 (5 μg/ml; normal goat IgG isotype control; R&D Systems) or RET (2 μg/ml; rabbit mAb IgG XP isotype control; Cell Signalling Technology) in PBS containing 1% BSA 1% donkey serum or 1% BSA and 0.3% Triton x-100, respectively, overnight at 4°C. Wells were washed three times with PBS and lung macrophages incubated with donkey antigoat IgG AF647 secondary antibody (1 μg/ml; Invitrogen) and THP-1 macrophages incubated with goat anti-rabbit IgG AF647 secondary antibody (1:1,000; Cell Signalling Technology) for 1 h at room temperature. Wells were washed three times with PBS and coverslips mounted onto slides using Prolong Gold Antifade reagent with DAPI overnight at room temperature in the dark. Images were collected on a Zeiss Axioimager.D2 upright microscope using a 20×/0.5 EC Plan-neofluar objective and captured using a Coolsnap HQ2 camera (Photometrics) through Micromanager software v1.4.23. Specific band pass filter sets for DAPI and Cy5 were used to prevent bleed through from one channel to the next. Images were then processed and analysed using Fiji ImageJ (http://imagej.net/Fiji/Downloads).

### ELISAs

IL-6, TNFα, IL-12p40, IL-10 (all DuoSet; R&D Systems), mouse NRTN (Stratech), and human NRTN (Universal Biologicals) were measured in supernatants, cell lysates, or BAL fluid by enzyme-linked immunosorbent assay (ELISA) according to the manufacturer's instructions.

### Gelatin degradation assay

Matrix degradation by primary lung macrophages was measured using the QCM Gelatin Invadopodia Assay (EMD Millipore) according to the manufacturer's instructions. Lung macrophages were plated at $1.5 \times 10^5$ in Nunc Labtek 8-well chamber slides (Thermo Fisher Scientific). Cell induced degradation of fluorescent gelatin was quantified as either the percent of degradation area per cell or the percentage of degradation area per total cell area. Data are displayed as a relative percentage normalised to poly(I:C), with poly(I:C) set to 100%. Analysis was performed according to manufactures instructions using Fiji ImageJ (http://imagej.net/Fiji/Downloads). DAPI and phalloidin signal was thresholded for high intensities, and then analysed as particles to provide a cell (nuclear) count or measurement of cell area. For quantification of the area of degradation Cy3-gelatin signal was thresholded for low intensities, then analysed as particles. Images were obtained using a Leica DM IL LED inverted microscope using a 4×/0.1 Hi-Plan objective and captured using a Leica DFC3000 G camera. Specific band pass filter sets for DAPI, FITC and Texas Red were used to prevent bleed through from one channel to the next.

### Statistics

GraphPad Prism version 7 was used for all statistical analysis. For multiple dataset analysis of parametric data, ANOVA with Tukey's correction was applied. For multiple dataset analysis of non-parametric data, a Kruskal–Wallis test with Dunn's multiple comparisons test was applied. To compare two datasets, an unpaired or paired $t$ test was applied. Data are presented as the mean ± SD. $P$-values < 0.05 were considered significant (*$P < 0.05$, **$P < 0.01$, ***$P < 0.001$).

## Supplementary Information

## Acknowledgements

T Hussell is supported by an investigator award from the Wellcome Trust (202865/Z/16/Z). This research was also supported by a pre-competitive open innovation award from AstraZeneca and GlaxoSmithKline that formed the Manchester Collaborative Centre for Inflammation Research. We acknowledge the Manchester Allergy, Respiratory and Thoracic Surgery Biobank for supporting this project. The Bioimaging Facility microscopes used in this study were purchased with grants from Biotechnology and Biological Sciences Research Council (BBSRC), Wellcome Trust, and the University of Manchester Strategic Fund. We specially thank Peter March, Roger Meadows, and Steven Marsden for their help with the microscopy.

### Author Contributions

E Connolly: formal analysis, validation, investigation, visualization, methodology, and writing—original draft.
DJ Morgan: formal analysis, investigation, methodology, and writing—review and editing.
M Franklin: formal analysis and investigation.
A Simpson: resources.
R Shah: resources.
OJ Brand: formal analysis and investigation.
CP Jagger: investigation and writing—review and editing.
J Casulli: visualization.
K Mohamed: investigation.
AM Grabiec: conceptualization, supervision, and methodology.
T Hussell: conceptualization, supervision, funding acquisition, writing—original draft, and project administration.

### Conflict of Interest Statement

The authors declare that they have no conflict of interest.

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
