## [Reviewer comments · Life Science Alliance]

Life Science Alliance

Neurturin regulates the lung resident macrophage inflammatory response to viral infection

Emma Connolly, David Morgan, Miriam Franklin, Angela Simpson, Rajesh Shah, Oliver Brand, Christopher Jagger, Joshua Casulli, Karishma Mohamed, Aleksander Grabiec, and Tracy Hussell
DOI: <https://doi.org/10.26508/lsa.202000780>

Corresponding author(s): Tracy Hussell, University of Manchester and Emma Connolly, University of Manchester

Review Timeline:

Submission Date:	2020-05-15
Editorial Decision:	2020-06-08
Revision Received:	2020-09-04
Editorial Decision:	2020-09-08
Revision Received:	2020-09-15
Accepted:	2020-09-22

Scientific Editor: Shachi Bhatt

Transaction Report:

June 8, 2020

Re: Life Science Alliance manuscript #LSA-2020-00780-T

Prof. Tracy Hussell
University of Manchester
Manchester Collaborative Centre for Inflammation Research
2nd floor, Core Technology Facility
Oxford Road
Manchester, Greater Manchester M13 9PT
United Kingdom

Dear Dr. Hussell,

Thank you for submitting your manuscript entitled "Neurturin regulates the lung resident macrophage inflammatory response to viral infection" to Life Science Alliance. The manuscript was assessed by expert reviewers, whose comments are appended to this letter.

The referees appreciate the findings and have noted a few minor revisions that need to be sorted out. You can use the link below to upload the revised version.

Thank you for this interesting contribution to Life Science Alliance. We are looking forward to receiving your revised manuscript.

Sincerely,

Reilly Lorenz

Editorial Office Life Science Alliance
Meyerhofstr. 1
69117 Heidelberg, Germany
t +49 6221 8891 414
e contact@life-science-alliance.org
www.life-science-alliance.org

B. MANUSCRIPT ORGANIZATION AND FORMATTING:

Reviewer #1 (Comments to the Authors (Required)):

In this paper, Connolly et al study the role of neurturin, RET and type I interferon in human lung macrophage function, presenting evidence that these pathways can regulate the production of MMPs and cytokines. The authors conclude that neurturin acts via RET in the context of type I IFN stimulation and that its effects may be anti-inflammatory, as well as altering the pattern of MMP production in the lung. Preliminary evidence is presented that these pathways may be altered in smokers and in human lung cancer. The data presented are generally clear and a variety of

macrophage sources have been explored under different conditions. Inevitably, when using human material, there is a limited amount of mechanistic information and some aspects of the study appear somewhat preliminary in nature. This applies in particular to the data from patients, where the implications for disease or smoking have not been pursued in any detail, while the experiments on MMPs do not address what the implications might be for function in vivo. As it stands, the work is more of a brief report than an in-depth exploration of neurturin and RET function in macrophages. Some specific comments:

- 1) The Discussion makes it clear that some of the issues discussed here have been addressed in neurturin^{-/-} mice. Although I recognise that the strength of the current work is to focus on humans, it would have been interesting to examine eg how the effects on MMP function seen in vitro might have been relevant in vivo.
- 2) In general, it is not particularly clear why the MMPs presented were the ones to be examined. Was this because other MMPs were not altered by neurturin etc? What are the specific implications of the changes in those MMPs that were studied?
- 3) Have the authors examined macrophages from other tissues to investigate whether the effects of the neurturin-RET pathway are selective to the lung?
- 4) The experiments in Figure 2 do not prove that the effects of neurturin on RET expression "require" type I IFN, but simply show additive or parallel effects. Some attempt to block IFN signalling would be needed to justify the authors' conclusion.
- 5) Figure 2J appears to be mislabelled, as I believe the bottom panels should be annotated as "IFN" and not "RET"?
- 6) Figures 4B and 4D, E do not show controls for neurturin alone (B) or AD80 alone (D, E)
- 7) Some data on MMP expression from the patient samples shown in Figure 6 would be appropriate

Reviewer #2 (Comments to the Authors (Required)):

The authors showed that NRTN can play a role as anti-inflammatory molecule through the inhibition of inflammatory cytokines and the expression of MMP2 and MMP9 by lung macrophages in context of viral infection.

The work is well performed, however some major remarks can be done:

In results part, the impact of NRTN on the MMP2 and MMP9 expression is confused, the authors have not clearly demonstrated what the exact role of NRTN is:

- 1) Line 164: the title is not correct, the authors have shown that NRTN participates in the downregulation of MMP9 and not in the upregulation.
- 2) In the paragraph of the MMP9 expression level induced by IFN γ (Fig EV4, D and E), IFN γ decreases the expression of MMP9 compared to the untreated condition. How the authors can explain that they didn't find the same result at mRNA level, Fig EV4,B?
- 3) Line 179: the main text "Interestingly, polyI:C or IFN γ significantly reduced MMP9 cellular expression and its release from lung macrophages (Fig EV4E and F)" is not correct as in the figure EV4E and F, the result is significant for IFN γ but not for the PolyI:C condition.
- 4) In Fig 3A/B the authors showed no significant difference of MMP2 at a protein level between stimulated lung macrophages with IFN γ +/- NRTN and significant differences at mRNA level (Fig EV4C), can the authors discuss the discrepancy?
- 5) Lines 195-197 and Figure 4C, D and E: What are the MMP1 and MMP3 levels in absence of NRTN?

Minor remark:

Neurturin is usually written "Nrtn" as a gene and "NRTN" as a protein, please correct in the text and also for writing the BEAS-2B cell line.

The authors have shown the NRTN expression in human A549 and BEAS-2B cell lines. They should confirm this result on human primary lung epithelial cells. It might be interesting to check also the production of NRTN by the NK cells as the authors focus on the process of viral infection. NK cells could be another source of NRTN in the lung and could contribute to the increase of GFRa2 expression by macrophages.

In conclusion, the experiments are clearly presented but the authors need to improve the result part and the discussion to explain more clearly the influence of NRTN on the MMP2 and MMP9 expression.

Dear editorial members

Thank you very much for providing the reviewers comments for our manuscript entitled "Neurturin regulates the lung resident macrophage inflammatory response to viral infection" (Ref number LSA-2020-00780-T). We are pleased with their general enthusiasm for our study and grateful for their positive and constructive feedback. We have addressed the reviewer's comments individually below, with their comments in black text and our response in red.

Reviewer #1

1) The Discussion makes it clear that some of the issues discussed here have been addressed in neurturin-/- mice. Although I recognise that the strength of the current work is to focus on humans, it would have been interesting to examine eg how the effects on MMP function seen in vitro might have been relevant in vivo.

We agree with the reviewer and to address this we are currently attempting to derive a novel floxed GFR α 2 mouse so that we can cross it with macrophages specific cre lines. Unfortunately, this takes considerable time and expense. In the process of generating the floxed strain we also produced a global GFR α 2 knockout strain. Unfortunately, this displayed significant developmental defects, were unable to breed effectively and so we had to terminate the line. We do not have the neurturin knockout mice in house and feel that a viral infection in these mice would be difficult to interpret, and they are likely not to survive the infection. Thus, we absolutely agree with the reviewer, but feel it is beyond the remit of the current manuscript.

2) In general, it is not particularly clear why the MMPs presented were the ones to be examined. Was this because other MMPs were not altered by neurturin etc? What are the specific implications of the changes in those MMPs that were studied?

It was interesting that the qPCR array indicated that NRTN may increase MMP2 at the mRNA level as it has previously been reported that RET activation enhances the production of this MMP, along with MMP9, and this has been included in the text at line 175. The results from the gelatin degradation assay in Figure 3F and G were not what we expected and therefore this led us to investigate whether NRTN could alter the production of other types of MMPs. MMP1 and MMP3 were chosen as they have also been linked to RET previously and have the ability to degrade gelatin. We have now included this in the text at line 201-205 to highlight why these MMPs were specifically chosen. We did not investigate the effect of NRTN on other MMPs, but in the future the macrophage specific GFR α 2 deletion model will be used to elucidate not only the role of GFR α 2, but also of some of the less well characterised MMPs in respiratory viral infections. As above, this mouse strain is currently a long way from being able to use. We have now added to the discussion the implications of NRTNs effect on MMPs from line 269-312.

3) Have the authors examined macrophages from other tissues to investigate whether the effects of the neurturin-RET pathway are selective to the lung?

Unfortunately, it is difficult to obtain macrophages from other human tissue sources. However, since GFR α 2 is also expressed by peripheral blood monocytes, we believe this pathway will be relevant to macrophages at other tissue sites and have amended the discussion to reflect this.

4) The experiments in Figure 2 do not prove that the effects of neurturin on RET expression "require" type I IFN, but simply show additive or parallel effects. Some attempt to block IFN signalling would be needed to justify the authors' conclusion.

We agree with the reviewer and have amended the text to reflect that type I IFN is not required but may be one of additional proteins that may also increase RET.

5) Figure 2J appears to be mislabelled, as I believe the bottom panels should be annotated as "IFN" and not "RET"?

Apologies, we have corrected this.

6) Figures 4B and 4D, E do not show controls for neurturin alone (B) or AD80 alone (D, E)

Unfortunately, due to the number of primary cells available we were not able to include neurturin alone in Figure 4B and we have not been able to receive subsequent human lung tissue samples at this time to address this. Although we cannot confirm the effect of neurturin alone on MMP3 expression in our primary human macrophages, we have been able to generate the data for AD80 alone and have now added this to Figure 4D and E.

7) Some data on MMP expression from the patient samples shown in Figure 6 would be appropriate

As requested we have now added data on MMP2 expression from the patient samples to Figure 6.

Reviewer #2

1) Line 164: the title is not correct, the authors have shown that NRTN participates in the downregulation of MMP9 and not in the upregulation.

We have amended the title accordingly.

2) In the paragraph of the MMP9 expression level induced by INF β (Fig EV4, D and E), INF β decreases the expression of MMP9 compared to the untreated condition. How the authors can explain that they didn't find the same result at mRNA level, Fig EV4,B?

It is now widely recognised that a change in mRNA levels do not necessarily equate to the same effect on protein and so protein analysis is more relevant. We have mentioned this in the discussion text at line 320.

3) Line 179: the main text "Interestingly, polyI:C or INF β significantly reduced MMP9 cellular expression and its release from lung macrophages (Fig EV4E and F)" is not correct as in the figure EV4E and F, the result is significant for INF β but not for the PolyI:C condition.

Though poly I:C shows a trend, we agree that significance is only obtained for IFN and have amended the text accordingly at line 185.

4) In Fig 3A/B the authors showed no significant difference of MMP2 at a protein level between stimulated lung macrophages with INF β +/- NRTN and significant differences at mRNA level (FigEV4C), can the authors discuss the discrepancy?

We believe that polyI:C activation may produce additional proteins by macrophages that are needed for NRTN to regulate MMP production, which are not induced by INF β . INF β + NRTN

decreases the pro-inflammatory cytokines IL-6 and IL-12 at the mRNA level, however polyI:C is needed to observe the decrease in pro-inflammatory cytokines by NRTN at the protein level. We included polyI:C stimulation, as well as IFN β stimulation, for the protein analysis of MMPs as we did not know whether this would be needed to detect functional differences at the protein level. We have now discussed the discrepancy in more depth in the discussion at line 313-319.

5) Lines 195-197 and Figure 4C, D and E: What is the MMP1 and MMP3 levels in absence of NRTN?

This data has now been added to Figure 4.

Minor remark:

Neurturin is usually written "Nrtn" as a gene and "NRTN" as a protein, please correct in the text and also for writing the BEAS-2B cell line.

We thank the reviewer for pointing this out and have amended the text throughout.

The authors have shown the NRTN expression in human A549 and BEAS-2B cell lines. They should confirm this result on human primary lung epithelial cells. It might be interesting to check also the production of NRTN by the NK cells as the authors focus on the process of viral infection. NK cells could be another source of NRTN in the lung and could contribute to the increase of GFRa2 expression by macrophages.

This is an interesting point. We have not observed, and it has never been reported that NK cells express NRTN. We did attempt to examine primary lung epithelial cells from lung resections but were unable to prevent outgrowth of fibroblasts. Therefore, we were unable to determine which cell type was contributing to NRTN production. As NRTN is secreted, immunohistochemistry produced inconclusive results. We have therefore included a section in the discussion specifying that epithelial cells may not be the only source of NRTN during viral infection of the lung.

In conclusion, the experiments are clearly presented but the authors need to improve the result part and the discussion to explain more clearly the influence of NRTN on the MMP2 and MMP9 expression.

We agree with the reviewer and have now provided more detailed explanations and discussions in the text at lines 181-214; 269-312 to make the influence of NRTN on MMP expression clearer to the reader.

We hope that the manuscript is now suitable for publication and look forward to hearing from you.

September 8, 2020

RE: Life Science Alliance Manuscript #LSA-2020-00780-TR

Prof. Tracy Hussell
University of Manchester
Manchester Collaborative Centre for Inflammation Research
2nd floor, Core Technology Facility
Oxford Road
Manchester, Greater Manchester M13 9PT
United Kingdom

Dear Dr. Hussell,

Thank you for submitting your revised manuscript entitled "Neurturin regulates the lung resident macrophage inflammatory response to viral infection". We would be happy to publish your paper in Life Science Alliance pending final revisions necessary to meet our formatting guidelines.

Along with the points listed below, please also address the following points:

- please add ORCID ID for corresponding author-you should have received instructions on how to do so
- please add a Running Title for your manuscript in our system
- please update your callout for Table S1; you have a callout for Table EV1, but it should be Table S1-page 18
- please make scale bars in Figure 3F more visible
- please edit the Figure 6 legends to reflect the figure panels (legends for Fig 6 D-F seem to be missing). Please also adjust the callouts in the manuscript text accordingly.

A. FINAL FILES:

-- High-resolution figure, supplementary figure and video files uploaded as individual files: See our

detailed guidelines for preparing your production-ready images, <http://www.life-science-alliance.org/authors>

B. MANUSCRIPT ORGANIZATION AND FORMATTING:

Sincerely,

Shachi Bhatt, Ph.D.
Executive Editor
Life Science Alliance

September 22, 2020

RE: Life Science Alliance Manuscript #LSA-2020-00780-TRR

Prof. Tracy Hussell
University of Manchester
Manchester Collaborative Centre for Inflammation Research
2nd floor, Core Technology Facility
Oxford Road
Manchester, Greater Manchester M13 9PT
United Kingdom

Dear Dr. Hussell,

Thank you for submitting your Research Article entitled "Neurturin regulates the lung resident macrophage inflammatory response to viral infection". It is a pleasure to let you know that your manuscript is now accepted for publication in Life Science Alliance.

The references are not currently formatted according to Life Science Alliance's publishing guidelines, but we have requested our production team to edit it on our side.

*****IMPORTANT:** If you will be unreachable at any time, please provide us with the email address of an alternate author. Failure to respond to routine queries may lead to unavoidable delays in publication.*******

DISTRIBUTION OF MATERIALS:

Congratulations on a very nice paper. I hope you found the review process to be constructive and are pleased with how the manuscript was handled editorially. We look forward to future exciting submissions from your lab.

Sincerely,

Shachi Bhatt, Ph.D.
Executive Editor
Life Science Alliance